# Transparent Conductive Electrodes Based on Graphene-Related Materials

**DOI:** 10.3390/mi10010013

**Published:** 2018-12-26

**Authors:** Yun Sung Woo

**Affiliations:** Department of Advanced Materials Application, Korea Polytechnics, Seoul 13122, Korea; yswoo@kopo.ac.kr; Tel.: +82-31-749-4085

**Keywords:** transparent conducting electrode, flexible electrode, graphene, optoelectronic device

## Abstract

Transparent conducting electrodes (TCEs) are the most important key component in photovoltaic and display technology. In particular, graphene has been considered as a viable substitute for indium tin oxide (ITO) due to its optical transparency, excellent electrical conductivity, and chemical stability. The outstanding mechanical strength of graphene also provides an opportunity to apply it as a flexible electrode in wearable electronic devices. At the early stage of the development, TCE films that were produced only with graphene or graphene oxide (GO) were mainly reported. However, since then, the hybrid structure of graphene or GO mixed with other TCE materials has been investigated to further improve TCE performance by complementing the shortcomings of each material. This review provides a summary of the fabrication technology and the performance of various TCE films prepared with graphene-related materials, including graphene that is grown by chemical vapor deposition (CVD) and GO or reduced GO (rGO) dispersed solution and their composite with other TCE materials, such as carbon nanotubes, metal nanowires, and other conductive organic/inorganic material. Finally, several representative applications of the graphene-based TCE films are introduced, including solar cells, organic light-emitting diodes (OLEDs), and electrochromic devices.

## 1. Introduction

Transparent conductive materials have been extensively used as essential components of optoelectronic devices, such as liquid crystal displays, touch panels, organic light-emitting diodes (OLEDs), and solar cells. Furthermore, the development of foldable or wearable displays and photoelectric devices has led to a need for electronic conductors that are transparent as well as stretchable. Owing to its relatively high electrical conductivity and transparency, indium tin oxide (ITO) is considered as a standard transparent electrode material for such devices. However, the high cost of raw materials, poor mechanical flexibility, and relatively high temperature of ITO deposition significantly limit the scope of its practical applications. For this reason, it has been attempted to use various kinds of nanoscale materials, such as carbon nanotubes (CNT), graphene, metal nanowires, metal nanogrids, and thin films as a replacement for ITO in transparent conducting electrodes (TCEs) [1,2,3,4,5,6,7,8,9,10,11,12,13,14,15,16,17,18]. Of these, CNT-based transparent electrodes showed the TCE performance, with a sheet resistance of 24 Ω·Sq^−1^ at 83% transmittance [18]. TCEs that are composed of randomly distributed metal nanowire networks have also been reported to have high optical transparency, low sheet resistance, and excellent mechanical flexibility [9,16,17]. However, silver or copper nanowires are easily damaged by moisture and external mechanical impact and their adhesion to the plastic substrate is poor [19,20,21,22,23]. Graphene-based electrodes have been investigated using the liquid suspension of graphene and macro-scale graphene synthesized via chemical vapor deposition (CVD) [12,24,25,26,27,28]. Liquid-based suspensions of graphene have an advantage in coating, as they would enable relatively low-cost methods of spin coating, roll-to-roll processing, and printing to be used. Additionally, CVD graphene has excellent physical properties, namely a sheet resistance of 30 Ω·Sq^−1^ at 90% transmittance, and thus is a promising candidate for TCE technology [12]. 

Despite the fact that TCEs based on graphene are easy to process, low in cost, and have excellent stability, they have a disadvantage in that the sheet resistance is larger than that of metal-based transparent electrodes that exhibit the same level of transparency [29]. Therefore, in recent years, an attempt has been made to complement the shortcomings and disadvantages of each material by constructing a hybrid structure of metallic nanostructure and graphene or graphene oxide (GO) or reduced GO (rGO) [3,30]. It has been reported that the performance of TCEs is improved by hybridizing organic and inorganic materials, such as ITO and poly(3,4-ethylenedioxythiophene) polystyrene sulfonate (PEDOT:PSS) with graphene [31,32]. 

This paper will review the fabrication methods of TCEs using graphene or GO (or rGO), which have been studied previously, and the optical, electrical, and mechanical properties with their limited application. Next, recent studies that have attempted to overcome the limitations of TCEs made with graphene or GO (or rGO) by introducing hybrid TCEs containing other materials are summarized. The performances of various hybrid TCEs that are based on graphene are summarized in Table 1. Lastly, various applications in optics and optoelectronics, especially in several newly emerging areas, such as electrochromic devices, are addressed along with their challenges and prospects in these fields.

## 2. Fabrication of Graphene-Based Transparent Conducting Electrodes (TCEs)

### 2.1. Chemical Vapor Deposition (CVD) Graphene-Based TCEs

CVD graphene is usually produced by flowing hydrocarbon gas onto a transition metal catalyst in a high-temperature furnace, which has been regarded as the most promising way to synthesize high-quality large-area graphene [12,63,64,65,66]. CVD graphene that is grown on transition metals, such as Ni, Cu, Pt, and Co, can be used as a TCE after transferring it onto the desired transparent substrate by removing the underlying metal [67]. 

In the first report on CVD graphene, multiple layers of graphene were grown on a Ni substrate via carbon dissolution and segregation using a Ni catalyst by a CVD process [48,68]. However, Ni possesses high carbon solubility, which makes it difficult to control the number of graphene layers. Thus, a mixture of monolayer graphene and multilayer graphene were formed on Ni foil. A breakthrough in CVD graphene has been achieved by developing synthetic ways of producing large-area monolayer graphene on a Cu foil using a roll-to-roll method [12,64,69,70,71,72,73,74]. Unlike Ni, Cu has a low carbon solubility, which makes it possible to grow monolayer graphene with a grain size of several centimeters on Cu foil using a mixture of methane and hydrogen gas at a high temperature of 1000°C as shown in Figure 1a–c and Figure 2; the sheet resistance was reported to be 125 Ω·sq^−1^ at 97.4% transparency [12]. It has also been demonstrated that Cu–Ni alloy can be used to produce monolayer and multilayer graphene using CVD with methane gas as precursor, since the carbon solubility can be controlled by adjusting the atomic fraction of Ni in Cu (Figure 1d) [39,75,76,77,78,79]. For example, Chen et al. presented the CVD synthesis of large-area, primarily bilayer, graphene on Cu–Ni foil by the use of a cold-wall reactor with methane and hydrogen as precursors [74]. Additionally, Cho et al. recently reported that extremely thin Cu–Ni alloy film could promote the formation of monolayer graphene, regardless of alloying contents by constraining the total amount of carbon that was absorbed into the film [78]. There have also been various attempts to directly grow graphene on a glass substrate [36,37,38,80]. Recently, it was reported by Sun et al. that large-area and uniform graphene film could be directly grown on glass substrate using catalyst-free atmospheric CVD (APCVD), with the resulting material presenting a sheet resistance of 370–510 Ω·sq^−1^ at a transmittance of 82% [36,38]. In comparison, annealing-based capping-metal catalyzed synthesis provides a fairy high quality of graphene. Xiong et al. showed that monolayer graphene that was grown on various dielectric substrates via rapid thermal process of substrate coated with amorphous carbon and Ni thin films exhibited a low sheet resistance of ~50 Ω·sq^−1^ at 95.8% transparency [37]. However, the processing temperature of 1100 °C was too high and not applicable to glass and plastic substrates.

In order to utilize CVD graphene grown on catalyst metal as a TCE, a transfer step is required to separate graphene film from the catalyst metal and move it to a transparent substrate, such as glass or polyethylene terephthalate (PET) [3,33,52,67,81,82,83,84,85]. Generally, a polymer-support layer is employed to protect the graphene from the external force during the transfer process. After coating the graphene onto the catalyst metal with a protective layer, such as polydimethylsiloxane (PDMS) or poly(methylmethacrylate) (PMMA), a metal catalyst, such as Cu or Ni, is etched away using a chemical etchant, such as HCl, HNO_3_, etc. The as-prepared polymer/graphene film after etching the metal catalyst is cleaned with deionized (DI) water and then transferred onto the target substrate. After removing the supporting polymer layer by organic solvent, only some polymer residues are left on the substrate for use as TCEs [67,80,82]. However, one of the disadvantages of this polymer-support transfer method is that polymer residues and defects are generated on graphene [52]. Therefore, various methods have been devised for transferring graphene without a support layer, which are called “polymer-free” methods as described in Figure 3; for example, a thermal release tape-assisted transfer, using a tape with specific adhesives that strongly adheres to substrates at room temperature while losing adhesion at high temperature, and a metal-assisted transfer method, using a metal as a protective layer instead of a polymer [83,85]. In particular, the thermal release tape-assisted transfer method is likely to be suitable for large-scale production because it can inherit the roll-to-roll (R2R) production process for graphene growth [62,63]. For instance, Bae et al. have successfully demonstrated the R2R production and coating of CVD graphene onto flexible substrate using a thermal release tape-assisted transfer method (Figure 2) [12]. Additionally, Lin et al. transferred the graphene to the substrate using “graphite holder”, in which monolayer graphene was etched and the etchant was pulled out to be replaced by mixed solvents when the solution was pulled out with the syringe [82]. Although the polymer-free transferred graphene exhibits superior electronic characteristics, the size and shape of the transferred graphene film is limited by the graphite holder. Furthermore, Wang et al. developed a unique “clean-lifting transfer” technique using a controllable electrostatic attraction force to transfer graphene film on various rigid or flexible substrates [81]. In this method, the graphene is attracted to the negatively charged target substrate by the electrostatic force during the transfer process. However, there was a problem with the “polymer-free” transfer method, in that the graphene was deformed easily by external force, such as liquid fluctuation, during metal etching because of the lack of a support layer. Therefore, the development of the “transfer-free” method is crucial for promoting the application of graphene, although the transfer process will still play an important role in the production of graphene devices before the “transfer-free” method becomes mature. 

### 2.2. Graphene Oxide-Based TCEs

Another potential way of producing graphene film on a large scale for TCEs is to deposit GO sheets on the desired substrate and then reduce them using thermal and chemical methods [45,86,87]. Since GO with hydrophilic properties is easily dispersed in water or other organic solvents, it has the advantage that GO solutions can be easily made into a large-area film using a liquid-based, cost-effective method. 

In general, GO is synthesized by oxidizing graphite with strong acids, followed by intercalation and exfoliation in water, which is represented by the Brodie, Staudenmaier, or Hummers method (Figure 4a) [88,89]. All three methods involve the oxidation of graphite to various levels [90,91]. The polar oxygen functional groups of GO that are produced by the oxidation process, that is, epoxy and hydroxyl group on the base plane and carboxyl group at the edge, cause it to be exfoliated and disperse well in water and other polar solvents with the assistance of ultrasonic agitation. The prepared GO suspension is then uniformly deposited on an arbitrary substrate while controlling the thickness of GO films for the subsequent TCE production [42,44]. Methods of coating the GO suspension with a film include spin coating, spray coating, dip coating, electrophoretic deposition, Langmuir–Blodgett (LB) assembly, etc. [42,45,92,93,94,95,96]. Spin coating and spray coating are the most convenient ways to deposit a thin film on a substrate from liquid suspension. In particular, spray coating has been proposed as a suitable method for production scale, owing to its fast, scalable, and easy operation [92,93]. Dip coating is also a popular way of coating a GO film on a rigid or flexible substrate, including the immersion of a substrate into GO suspension, the draining of remaining suspension, and the drying of a substrate [94]. However, it was found to be very hard to avoid the partial aggregation and wrinkling or folding of the GO sheet during the spray, spin, or dip coating due to the high flexibility of the sheet. Electrophoretic deposition is performed through the migration of GO sheets in a suspension toward the positive electrode when a direct current (DC) voltage is applied [95]. Despite the many advantages of this technique in film preparation, such as high deposition rate, thickness controllability, and convenience in scaling-up, electrophoretic deposition is limited by the fact that only conductive substrates, such as ITO-coated glass, Al, Ni, and stainless steel are applicable for TCE fabrication. LB assembly is a sophisticated method which allows continuous and uniform film to be deposited on an arbitrary substrate, in which GO sheets floating on water are compressed by LB trough until the desired surface pressure is reached to realize the uniform deposition of GO sheets [96,97].

Since GO thin film is electrically insulating, it is necessary to reduce it using the thermal and chemical method to recover its conductivity for TCE application. Chemical and thermal reduction is the most common way of reducing GO thin film [97]. In the chemical reduction process, various inorganic and organic reducing agents are applied, such as phenyl hydrazine, hydrazine hydrate, sodium borohydride, ascorbic acid, glucose, hydroxylamine, hydroquinone, pyrrole, amino acids, strongly alkaline solution, and urea [99,100]. However, the chemical reduction is not sufficient to completely recover graphene from GO, and it leaves a substantial amount of residual functionality of epoxy and hydroxyl and carbonyl groups. Thermal reduction by annealing is generally regarded as a more efficient way to reduce GO than chemical reduction [101,102]. Many researchers have reported that the conductivity of GO thin film increases with annealing temperature in a vacuum or reducing atmosphere. In a recent report, rGO thin film exhibited a conductivity of up to 10^4^ S∙cm^−1^ and a transparency on the level of 90% after annealing at 1100 °C under Ar or N gas flow (Figure 3b–d) [43]. However, since most thermal reduction is carried out at high temperature above 1000 °C for a relatively long time, this approach is not applicable for plastic substrates for flexible TCEs. Other methods have been attempted to reduce GO, such as microwave-assisted heating and the removal of the functional group using a photocatalyst, such as TiO_2_ [43,103].

### 2.3. Chemical Doping of Graphene

Although the graphene films that were prepared using the methods described in Section 2.1 and Section 2.2 possess excellent electrical properties and high transparency, their sheet resistance is still too high for the sheets to be used as TCEs. One approach to reduce the sheet resistance of graphene film is post- chemical doping of graphene after transferring it to the desired substrate. The chemical doping of graphene, achieved using chemical species, is classified with surface transfer doping and substitutional doping [104,105]. Surface doping is achieved by charge transfer between graphene and a dopant that is absorbed on the surface of graphene. Graphene can be p-type or n-type doped via chemical doping, depending on the relative position of density of states (DOS) of the highest occupied molecular orbital (HOMO) and lowest unoccupied molecular orbital (LUMO) of the dopant and the Fermi level of graphene. If the HOMO of a dopant is above the Fermi level of graphene, hole transfers take place from the dopant to the graphene, inducing the p-type doping of graphene. If the LUMO of a dopant is below the Fermi level of graphene, then electron transfer takes place from the dopant to the graphene, inducing the n-type doping of graphene. 

The resistance of graphene is significantly decreased by charge transfer leading to the p-type doping of graphene when the graphene is exposed to HNO_3_, NO_2_, SOBr_2_, Br_2_, and I_2_ [105,106,107,108]. Karsy et al. demonstrated that an eight-stacked layer of graphene which was interlayer-doped with HNO_3_ exhibited a sheet resistance of 90 Ω·sq^−1^ at a transmittance of 80% [106]. As described in Figure 5a, through the layer-by-layer doping of four layers of graphene with HNO_3_, a sheet resistance of graphene of 80 Ω·sq^−1^ was achieved at a transmittance of 90% [108]. Redox dopants, such as AuCl_3_ and AgNO_3_, were also found to lower the sheet resistance of graphene TCEs when graphene was immersed in AuCl_3_ or AgNO_3_ solution and Au^3+^ was reduced to form Au nanoparticles on graphene by charge transfer from graphene [109,110]. Surface-adsorbed molecules with electron withdrawing groups can induce a p-type doping effect in graphene. Tetrafluoro-tetra-cyanoquiondimethane (F4-TCNQ) is a strong electron acceptor that is widely used to improve the performance of graphene TCEs for various devices [111,112,113]. For instance, by using the local density functional theory, Pinto et al. demonstrated that a charge transfer of 0.3 holes/molecule occurs between graphene and F4-TCNQ, which is in agreement with the experimental findings on F4-TCNQ [112]. Graphene can be doped with electrons, that is, n-type doping, via donors such as potassium, ethanol, and ammonia [41,114,115]. Additionally, polymers with amine groups can be used to produce electron-doped graphene. For example, Jo et al. demonstrated a stable and strong n-type doping method with pentaethylene hexamine (PEHA), which reduced the sheet resistance of graphene by up to ~400% compared to pristine graphene [116]. The dual-side n-doping of graphene with diethylenetriamine (DETA) on the top and amine-functionalized self-assembled monolayers (SAMs) at the bottom has been developed to enhance the conductivity of graphene. This method resulted in a sheet resistance as low as ~86 Ω·sq^−1^ with a transmittance of ~96% (Figure 5b–e) [41]. 

## 3. TCEs of Graphene-Related Materials Hybridized with Other Materials

Combining graphene and other conductive materials can overcome the drawbacks of each individual material. Based on this concept, the hybridization of graphene with various materials, including metal nanowires, carbon nanotubes, and conductive organic and inorganic materials, has been attempted to enhance the conductivity and the optical characteristics of TCEs.

### 3.1. Hybridization of Graphene with Carbon Nanotubes

CNTs have delivered high axial carrier motilities, making them an obvious choice for use as TCEs. However, TCE films consisting of CNTs have a spiderweb-like network structure with many voids between nanotube bundles; the presence of these voids contributes to the high transparency of CNT films, but restrain the conductance of the films. Meanwhile, CVD graphene presents a fundamental limitation in transmittance and sheet resistance due to its polycrystalline structure, making it difficult to compete with ITO. 

To overcome these shortcomings of each substance, a composite that is based on polycrystalline CVD graphene and a subpercolating network of nanowires has been demonstrated, in which nanowires provide the number of electronic pathways to bridge the percolating bottleneck, such as high resistance grain boundary, resulting in reduced resistance while maintaining high transmittance [46,47,51,117]. Kim et al. synthesized a graphene hybrid film by growing graphene using thermal chemical vapor deposition on Cu foil that was coated with single-walled CNTs (SWNTs). The SWNT/graphene hybrid film exhibited superior TCE properties, with a sheet resistance of 300 Ω·sq^−1^ and transmittance of 96.4%, as compared to graphene spin-coated with SWCNTs, which had a sheet resistance of 1100 Ω·sq^−1^ and a transmittance of 96.2%. This is presumably due to the low contact resistance between graphene and SWNTs in the hybrid film [47]. Another type of CNT/graphene hybrid film, which has been named as rebar graphene sheets synthesized by annealing the functionalized multi-walled CNT (MWNT) on Cu foil without an exogenous carbon source, has been reported to have ~95.8% transmittance with a sheet resistance of ~600 Ω·sq^−1^ [118]. Kholmavnov et al. also reported that when CVD graphene is on the top of the MWNT sheet layer (the “G/MWNT” configuration), it significantly modified the MWNT sheet, giving rise to better electrical and optical properties than the reversed structure of the “MWNT/G” configuration [119]. 

There have also been several attempts to form a nanocomposite comprised of carbon nanotubes and rGO or chemically converted GO (CCG) as shown in Figure 6. The rGO/MWNT double layer was prepared by the sequential electrostatic adsorption of negatively charged GO and positively charged MWNTs on the substrate, followed by the reduction of GO in hydrazine solutions and annealing under an argon atmosphere. The sheet resistance of the rGO/MWNT thin films had the lowest value of 151 kΩ·sq^−1^ for a 60 μg/mL concentration of aminated MWNT, with a transparency of 93% at a wavelength of 550 nm [48]. Another approach to combine CCG and CNT is to produce hybrid suspension of CCG and CNTs (called G-CNT), as reported by Tung et al. [49]. The stable G-CNT dispersion in hydrazine was readily deposited on a variety of substrates by spin-coating and subsequently heated to 150 °C to remove excess solvents. The G-CNT film had an optical transmittance of 92% and a sheet resistance of 636 Ω·sq^−1^, which is two orders of magnitude lower than that of the analogous vapor-reduced GO film. This vast improvement in sheet resistance for the G-CNT film is presumably due to the formation of an extended conjugated network with individual CNTs bridging the gap between graphene sheets. 

The method of LB assembly was also employed to deposit GO and SWNT in a layer-by-layer manner (Figure 6a). Zheng et al. prepared ultra-large GO sheet/SWNT hybrid films using the LB assembly technique, in which COOH-functionalized SWNTs were crucial for the successful deposition of SWNT layers [50]. The GO/SWNT hybrid film on the substrate was subsequently thermally reduced by heating at 1100 °C to achieve graphitization. A remarkable transmittance, exceeding 90% at a wavelength of 550 nm, was delivered by the 0.5–1.5 bilayer hybrid films and decreased gradually with increasing numbers of bilayers. However, the sheet resistance of the 1.5-bilayer hybrid film was ~600 Ω·sq^−1^, requiring further improvement by additional acid treatment or doping. 

### 3.2. Hybridization of Graphene with Metal Nanostructure

TCEs based on metallic nanostructures, such as metallic nanowires and patterned metal grids, have attracted much attention due to their promising properties of low sheet resistance, high optical transparency, and excellent mechanical durability [17,29]. However, when subjected to conditions of high temperature and current, metallic nanowire networks can experience early failure rates that are caused by the electromigration process and can be destroyed by chemical surface reaction [56,120]. TCEs composed of metallic nanowire networks have the limitation of increasing the electrical conductivity due to the junction resistance between the individual nanowires [121]. Recently, the hybridization of metallic nanostructures and graphene has been devised with the hope that the graphene will complement these drawbacks of TCE of metallic nanostructures, schematically described in Figure 7 [3,16,57,122]. Zhu et al. developed a graphene/metal grid hybrid electrode that was placed onto PET film whose sheet resistance was ~20 Ω·sq^−1^ at 90% transmittance [123]. The graphene/metal grid hybrid electrode was fabricated by the following sequence. First, metal grids were formed on a transparent substrate, such as PET film. Subsequently, a graphene film grown on Cu foil was transferred to the top of the grid via the wet transfer method and the sacrificial PMMA layer was removed to form the final graphene/metal grid hybrid transparent electrodes. Additionally, high-performance dye-sensitized solar cells fabricated using a hybrid TCE of Pt or Ni grids covered by graphene have been demonstrated, whose efficiencies were comparable to that of the oxide-based transparent electrode [124]. The superior optical and mechanical properties of these graphene/metal grid hybrid electrodes compared to conventional TCEs has motivated the study of hybrid electrodes with various types of graphene and metal nanostructures [60,125]. 

TCEs composed of a random network of metal nanowires, such as silver or copper nanowires, have the advantage that they can be manufactured in an inexpensive roll-to-roll process while maintaining the high conductivity of the metal [32,51,53,54]. As shown in Figure 8, a hybrid structure employing CVD graphene and a network of silver nanowires has shown a very low sheet resistance of 22 Ω·sq^−1^ at 88% transmittance with excellent stability, and its sheet resistance was stabilized to 13 Ω·sq^−1^, even after 4 months [54]. The co-percolating conduction model demonstrates that these superior TCE properties of hybrid structure of CVD graphene and a network of metal nanowires are due to the fact that the high-resistance grain boundaries in graphene are bridged by the silver nanowires and the junction between the nanowires are bridged by graphene, and consequently a low sheet resistance was possibly accomplished, even at moderate nanowire densities. Furthermore, this hybrid TCE of graphene and metal nanowires exhibited multiple functionalities, such as robust stability against electrical breakdown and oxidation and superb flexibility [55,77,126,127]. The hybrid silver nanowire and graphene electrode showed the ultimate flexibility without experiencing a significant change of sheet resistance for bending radii of curvature as small as 3.7 µm with a strain of ~27%. Lee et al. presented an inorganic light-emitting diode (ILED) on a soft contact lens that was fabricated with this hybrid metal nanowire and graphene TCE, with the results suggesting a substantial promise for future flexible and wearable electronics and implantable biosensor devices [77]. Moreover, Metha et al. fabricated graphene-encapsulated copper nanowires, whose electrical and thermal conductivity outperformed those of uncoated copper nanowires using a low-temperature plasma-enhanced CVD, with the results suggesting that graphene-encapsulated copper nanowires can be adopted for TCEs in air-stable flexible device applications [55]. On the other hand, Deng et al. demonstrated continuous R2R production of TCEs based on a metal nanowire network that was fully encapsulated between a graphene monolayer and plastic. This low cost and scalable manufacturing method of graphene/metal nanowire hybrid TCEs is expected to accelerate its application to various fields in industry [54].

### 3.3. Hybridization of Graphene Oxide with Metal Nanostructure

Hybrid structures of metal nanowires and GO (or rGO) for the production of high-performance transparent electrodes have developed since the low-cost and large-scalable solution process has made it possible to deposit GO (or rGO) film on a plastic substrate. Typically, hybrid TCEs of silver or copper nanowires and GO (or rGO) have been prepared by coating GO (or rGO) onto the silver nanowire film using the dip, spin, and spray coating method [86]. 

As mentioned above, the issue of the long-term stability of metal nanowire film makes it difficult to use in practical TCEs. However, GO (or rGO)-coated silver nanowire films exhibited highly enhanced long-term stability due to the excellent gas barrier properties of the GO (or rGO) passivation layer on metal nanowire film [78,127,128,129]. Ahn et al. reported that the sheet resistance of silver nanowire/rGO film was slightly increased, by less than 50%, even at 70 °C and 70% relative humidity (RH) for eight days, while the silver nanowire film showed increased sheet resistance, more than 300% [126]. Additionally, it has been shown that hydrophilic GO nanosheet can be used as a novel adhesive overcoating layer on hydrophilic silver nanowire/PET film that tightly holds the silver nanowires and reduces the sheet resistance in Figure 9 [128,130]. This GO/silver nanowire hybrid TCE also exhibited excellent bendability, showing an almost constant sheet resistance through over 10,000 bending cycles with a ~2 mm curvature radius. 

It has also been reported that a high-quality copper/rGO core/shell nanowire could be obtained by wrapping the GOover the surface of the copper nanowire and subsequent mild thermal annealing. These ultrathin core-shell nanowires produced high-performance TCEs with excellent optical and electrical properties, that is, with a sheet resistance of ~28 Ω·sq^−1^ and a haze of ~2% at a transmittance of ~90%, as reported by Duo et al. [131]. It has also been demonstrated that a film composed of rGO assembled onto copper nanowire film has improved electrical conductivity, oxidation resistance, substrate adhesion, and stability in harsh environments [132]. In particular, an electrochromic device employing the rGO/copper nanowire hybrid TCEs showed reversible coloration/bleaching properties, which cannot be obtained using pure copper nanowire TCEs, since these nanowires form copper hexacyanoferrate compounds during the electrochemical bleaching process, while the rGO protects the copper nanowire from reacting with the harsh solution used for the deposition of electrochromic material [133].

### 3.4. Hybridization of Graphene with Conducting Polymer

PEDOT:PSS/graphene composite has been fabricated using various methods to give improved performance in chemical and electrical properties [31,58,59,60,134].

Jo et al. produced a stable aqueous suspension of rGO nanosheet through the chemical reduction of GO in the presence of PEDOT:PSS [59]. The resultant rGO/PEDOT:PSS suspension yielded a hybrid TCE film with a high conductivity of 2.3 Ω·sq^−1^ with a transmission of 80%. An easy, low cost mass production of PEDOT:PSS/graphene composite has also been developed through the in situ polymerization of PEDOT in the presence of rGO and high-molecular PSS [31]. Since the rGO was used, no additional reduction processes of graphene were required. Liu et al. employed electrochemically exfoliated graphene to prepare the hybrid ink of PEDOT:PSS and graphene, since the electrochemical exfoliation of graphite produces high-quality graphene at a bulk scale (Figure 10) [58]. In order to disperse exfoliated graphene at higher concentrations, PH1000 (Heraeus Clevious, USA) was selected as a surfactant due to its conjugated aromatic chains that can strongly anchor onto the graphene surface via π–π interactions. Subsequently, the TCE films were prepared by the spray-coating method, and their sheet resistance was measured to be 500 Ω·sq^−1^ at a transmission of 80% after applying 100 cycles of spray-coating. 

On the other hand, graphene/PEDOT:PSS bilayers have been utilized for the purpose of using PEDOT:PSS as a new supporting layer for the transfer of CVD graphene film without a removal process [124]. An important advantage of this approach is that PEDOT:PSS acts as an effective dopant for CVD graphene film, which exhibits a sheet resistance of 80 ± 4 Ω·sq^−1^ with excellent stability in air. This graphene/PEDOT:PSS bilayer exhibited a high transparency, with a transparency decrease of only 1% being caused by adding a PEDOT:PSS layer.

### 3.5. Hybridization of Graphene with Oxide

There are not many reports on the fabrication of hybrid metal oxide/graphene TCEs when compared to hybrids using organic materials or nanomaterials with graphene. 

Transparent conductive oxides, such as ITO and aluminum-doped zinc oxide (AZO), are widely used in electrodes, however the rigidity of these materials has limited their use as flexible electrodes. There have been attempts to improve the mechanical properties of ITO by fabricating a hybrid electrode using graphene and ITO [61,62]. Liu et al. demonstrated that the graphene/ITO hybrid electrodes showed a resistance change (ΔR/R_0_) of 17.78 after 20% tensile strain; meanwhile, the (non-hybrid) ITO electrode showed a resistance change of 125.91. When the bending radius was 0.1 cm, the resistance change of the ITO electrode fell to ~115.51, while that of the graphene/ITO hybrid electrode fell to ~11.65. These results showed the benefits of the graphene/ITO hybrid electrode over the ITO electrode in terms of mechanical flexibility [61]. Another approach for the hybridization of graphene with ITO was the uniform dispersal of ITO nanoparticles with a size of 25–35 nm on CVD graphene, which is synthesized by the immersion of graphene into aqueous ITO sol-gel. The ITO nanoparticle-decorated graphene exhibited a decrease in sheet resistance of about 28.2% relative to that of CVD graphene, owing to the electron doping of graphene that is induced by the ITO nanoparticles [62]. 

As shown in Figure 11, a multilayered electrode in which graphene is sandwiched between metal oxide has been demonstrated to have high electrical stability and optical transparency [69]. Since the coating of graphene with metal oxide prevents the desorption of chemical dopants, graphene that is coated with a 60-nm-thick layer of WO_3_ showed a lower resistance and slower degradation rate than pristine graphene. Furthermore, the optical transmittance of this WO_3_-coated graphene could be enhanced with the addition of a metal oxide layer between the graphene and the glass, which satisfies the zero reflection condition. As a result, the optimal multilayered structure of TiO_2_ (62 nm)/graphene (3 layers)/WO_3_ (60 nm) on glass showed a transmittance of ~90%, which is same as the highest transmittance of glass/ITO.

## 4. Application of Graphene-Based TCEs

The excellent performance of various graphene-based TCEs give graphene a realistic chance of becoming competitive in the production of transparent and bendable device technologies. In particular, the combination of high chemical and thermal stability, high stretchability, and low contact resistance with organic materials offers tremendous advantages for using graphene for TCEs in organic electronic devices, such as solar cells, OLEDs, touch screens, field effect transistors, sensors, and electrochromic devices [70,135]. 

### 4.1. Solar Cells

A low-cost, exfoliated graphene oxide followed by thermal reduction was first studied for application as window electrodes in solid-state dye-sensitized solar cell [45]. Eda et al. utilized reduced and doped GO thin films as the cathode in optovoltaics (OPVs), which were fabricated by spin coating a layer of PEDOT:PSS on top of a rGO film and subsequently depositing a poly(3-hexylithiophene) (P3HT) and phenyl-C61-bytyric acid methyl ester (PCBM) nanocomposite layer [136]. Additionally, a rGO film with 1100ºC thermal annealing has been utilized as an anode in a dye-sensitized solid solar cell based on spiro-OMeTAD and porous TiO_2_, however the short-circuit photocurrent density (I_sc_) and efficiency of the graphene-based cell was somewhat lower when compared to an fluorine-doped tin oxide (FTO)-based cell, possibly due to the series resistance of the device, the relatively lower transmittance of the electrode, and the electronic interfacial change. 

Subsequently, an organic solar cell fabricated with TCEs based on CVD graphene with a low sheet resistance has demonstrated excellent performance with an enhanced power conversion efficiency (PCE) [33,137]. Since CVD graphene offers high conductivity when compared to rGO, it offers a great advantage for the fabrication of OPV devices. A multilayer graphene (MLG) film with a relatively low sheet resistance of 374 Ω·sq^−1^ at 84.2% transparency was obtained, and the MLG) film-based solar cell with a P3HT:PCBM blend as the active layer exhibited a PCE of 1.17% [137]. An inverted structural solar cell with AZO at the bottom as cathodes, molybdenum-oxide/graphene on top as anode, and P3HT:PCBM as an active layer exhibited a PCE of 2.2%. 

Various hybrid TCEs using graphene and a highly conductive material have been employed as a window electrode in the fabrication of OPVs [49,55,60,127,134]. Lee et al. reported that a CVD graphene/PEDOT:PSS bilayer could provide the highest PCE of 5.5% and a fill factor (FF) of 0.67, which is even higher than what is obtainable with the best ITO device (Figure 12) [60]. Additionally, rGO-coated silver nanowire film with excellent thermal and chemical stabilities has been used for the anode layers in bulk heterojunction polymer solar cell [127]. Under illumination, this showed an open-circuit voltage (V_oc_) of 0.49 V, a short-circuit current density (J_sc_) of 6.38 mA∙cm^−2^, and an FF of 32.91, resulting in a PEC of 1.03%. Furthermore, a solution-based nanocomposite that is comprised of chemically converted graphene and carbon nanotubes has been used as a platform for the fabrication of P3HT:PCBM photovoltaic (PV) devices. After spin-coating a mixture of graphene and CNT, a thin buffer layer of PEDOT:PSS and P3HT:PCBM with a 1:1 weight ratio was coated on glass [49]. Finally, the Al and Ca were evaporated as the reflective cathode. For these PV devices, a PCE of 0.85% was measured under an illumination of AM 1.5 G and the values of J_sc_, V_oc_, and FF were 3.47 mA∙cm^−2^, 0.583 V, and 42.1%, respectively. These relatively low values of J_sc_ and FF, which are detrimental to PCE, were likely due to poor contact at the interface between the graphene/CNT composite and polymer. 

### 4.2. Organic Light-Emitting Diodes (OLEDs)

At first, a basic OLED structure of anode/PEDOT:PSS/*N*,*N*′-di-1-naphthyl-*N*,*N*′-diphenyl-1,1′-biphenyl-4,4′diamine (NPD)/tris(8-hydroxyquinoline) aluminum (Alq_3_)/LiF/Al was adopted to investigate the performance when a graphene film was used as the transparent electrode [26]. The graphene electrodes were deposited on quartz slides by spin-coating water-based dispersion of functionalized graphene and reduced by high-temperature vacuum annealing; the sheet resistance of the electrodes was ~800 Ω·sq^−1^ at a transmittance of 82% at 550 nm. An OLED with graphene anode exhibited a turn-on voltage of 4.5 V and it reached a luminance of 300 cd∙m^−2^ at 11.7 V, which is slightly higher than the values for ITO. A sky-blue phosphorescent OLED with multilayered CVD graphene anode, consisting of graphene (2–3 nm)/1,1-bis[(di-4-tolylamino)phenyl]cyclohexane (TAPC) (30 nm)/1,4,5,8,9,11-hexaazatriphenylene hexacarbonitrile (HAT-CN)(10 nm)/TAPC(30 nm)/HAT-CN(10 nm)/TAPC(30nm)/4,40,400-tri(N-carbazolyl)triphenylamine:iridium(III)bis[(4,6-difluorophenyl)–pyridinato-N,C20] picolinate (TCTA:FIrpic)(5 nm)/2,6-bis[30-(N-carbazole)phenyl] pyridine:iridium (III) bis[(4,6-difluorophenyl)-pyridinato-N,C20] picolinate (DCzPPy:FIrpic) (5 nm)/1,3-bis(3,5-dipyrid-3-yl-phenyl)benzene(BmPyPB) (40 nm)/lithium fluoride (LiF) (1 nm)/aluminum (Al) (100 nm), showed an electron quantum efficiency (EQE) of 15.6% and a power efficiency (PE) of 24.1 lm/W, comparable to the EQE of 18.5% and PE of 28.5 lm/W obtained with an ITO-based OLED [34]. 

Flexible OLEDs that were fabricated with graphene-based anode shown in Figure 13 have also been demonstrated in several recent reports [70,130,137,138]. A hybrid graphene/silver nanowire/polymer electrode prepared on PET provided not only stable resistance against air exposure, but also superior flexibility with the bending cycles [138]. The optimal OLED device based on a hybrid graphene/silver nanowire/polymer electrode presented the best performance due to its higher conductivity and light transmittance [130]. A GO-soldered silver nanowire network was also provided as an anode of the fully stretchable polymer light-emitting diode (PLED) with the help of GO solder, which improves the stretchability of the silver nanowire network. 

### 4.3. Electrochromic Devices

Electrochromic devices are widely used in display devices, including smart windows and mirrors, for improving indoor energy efficiency or personal visual comfort. 

Recently, rigid electrochromic devices have advanced, and they now offer flexibility, stretchability, and foldability for deformable devices (Figure 14) [119,133,140,141]. Polat et al. demonstrated a flexible electrochromic device using multilayer graphene, which offers key requirements for practical application, that is, high-contrast optical modulation over a broad spectrum and good electrical conductivity and mechanical flexibility [141]. They showed that the optical transmittance of MLG can be controlled by electrostatic doping via the reversible intercalation of charges into the graphene layers, making it possible to fabricate reflective/transmissive multipixel electrochromic devices. 

Smart windows that are based on electrochromic devices were demonstrated by Kim et al., who investigated the electrochemical and electrical characteristics of the PEDOT:PSS polymer-based electrochromic devices s of the different number of graphene layers used as electrodes [142]. The four-layered graphene electrode showed the best electrochemical behaviors, with a fast optical change response of less than 1 s from the dark to the transparent state and 500 ms from the transparent to the dark state and a low bias of ± 2.5 V for the maximum contrast ratio. 

### 4.4. Transparent Heaters 

High performance, transparent, and flexible heaters have been fabricated using graphene-based electrothermal films that are suitable for automobile defogging or deicing systems and heatable smart windows, as shown in Figure 15 [128,143,144]. A multiple-stacked CVD graphene film on PET, being interlayer-doped with AuCl_3_-CH_3_NO_2_ and HNO_3_ to obtain a low sheet resistance, was provided for flexible heaters, and it was mechanically stable after bending 1000 times with 1.1% strain [142]. Additionally, GO film spin-coated on quartz or polyimide (PI) has also exhibited high transparency and good heating effects. 

A new strategy of reduced large-size graphene oxide (rLGO)/silver nanowirehybrid film has been reported to design high-performance transparent film heaters. The thin rLGO provided a protective effect for the silver nanowire network against oxidation as well as the low sheet resistance of rLGO [128].

## 5. Summary and Conclusions

New TCE materials, including carbon nanotubes, metal grids and nanowires, conductive polymers, and recently graphene, have appeared over the past decades in many fields of application. 

In particular, these new materials offer prospective advantages of flexibility, bendability, and even stretchability, enabling them to be applied as TCEs in wearable optoelectronic devices and displays. Among them, graphene has attracted special attention due to its superior electrical conductivity and optical transmission when compared to other TCE materials. Additionally, the excellent moisture barrier and mechanical flexibility of graphene allows it to be hybridized with other TCE materials to further improve the properties of TCE films.

In the past decades, a number of methods for synthesizing high-quality graphene at low cost have been extensively studied, and the development of a large-area synthesis and transfer method using a roll-to-roll process has been a viable step in the practical application of graphene TCEs. Despite the tremendous advances in the graphene synthesis process and quality improvement, TCE films that are based on graphene still have issues that need to be improved, such as poor adhesion to substrates, low abrasion, and poor electrical conductivity as compared to conventional ITO.

## Figures and Tables

**Figure 1 micromachines-10-00013-f001:**
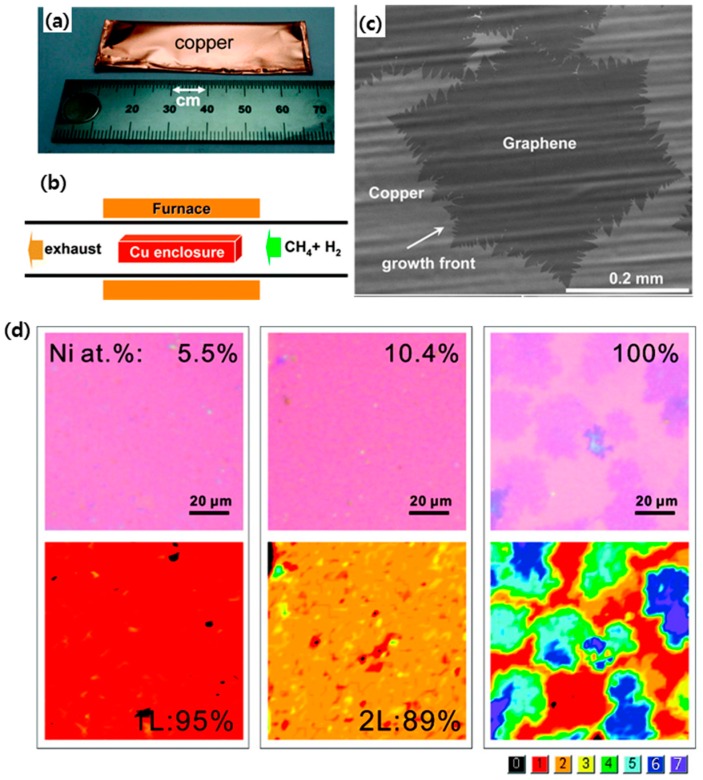
(**a**) Copper foil enclosure prior to insertion in the furnace. (**b**) Schematic of the chemical vapor deposition (CVD) system for graphene on copper. (**c**) SEM image of graphene on copper grown by CVD. Graphene grown at 1035 °C on Cu at an average growth rate of ~6 μm/min. Reproduced with the permission of Reference [65], Copyright 2011. American Chemical Society (**d**) Morphology and layer distribution of various few layers graphene segregated from Cu-Ni alloy at 900 °C after transfer to 300 nm SiO_2_/Si substrate. Reproduced with the permission of Reference [76]. Copyright 2011, American Chemical Society.

**Figure 2 micromachines-10-00013-f002:**
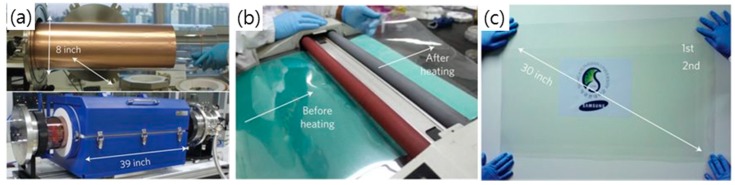
(**a**) Copper foil wrapping around a 7.5-inch quartz tube to be inserted into a 8-inch quartz reactor. The lower image shows the stage in which the copper foil reacts with CH_4_ and H_2_ gases at high temperatures. (**b**) Roll-to-roll transfer of graphene films from a thermal release tape to a polyethylene terephthalate (PET) film at 120 °C. (**c**) A transparent ultralarge-area graphene film transferred on a 35-inch PET sheet. Reproduced with the permission of Reference [12]. Copyright 2010, Nature Publishing Group.

**Figure 3 micromachines-10-00013-f003:**
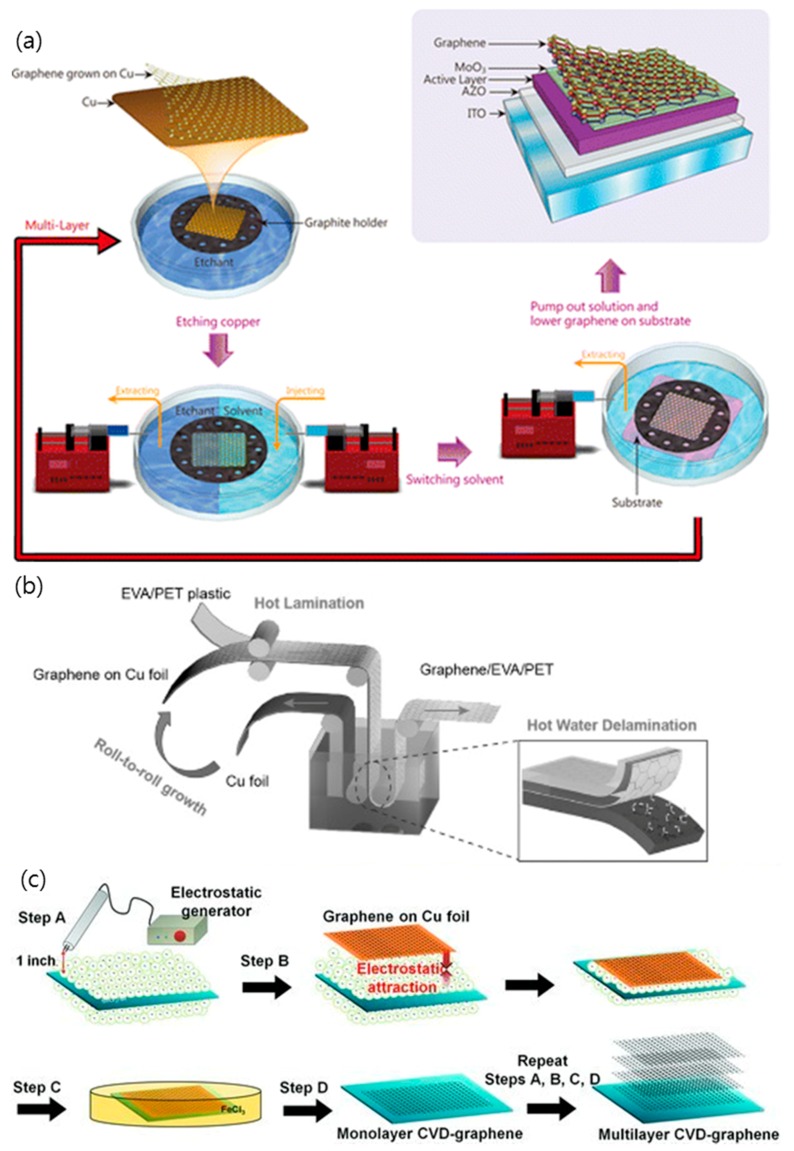
Schematic illustration of (**a**) the polymer-free transfer process, (**b**) the roll-to-roll delamination of copper and graphene onto ethylene-vinyl acetate (EVA)/polyethylene terephthalate (PET) substrate, and (**c**) the clean-lifting transfer process of as-grown graphene on copper foil onto a substrate. Reproduced with the permission of Reference [33,35,81]. Copyright 2014, American Chemical Society. Copyright 2015, 2013, Wiley-VCH.

**Figure 4 micromachines-10-00013-f004:**
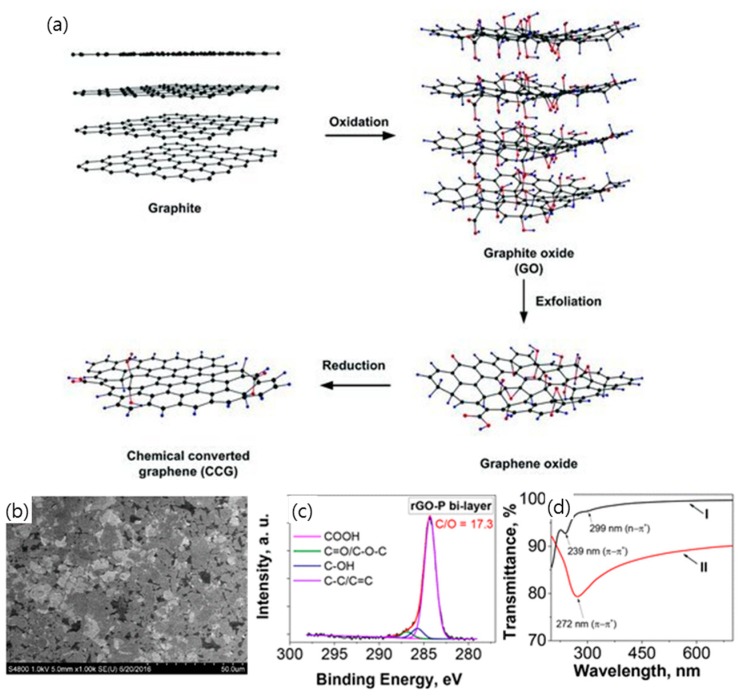
(**a**) Preparation of graphene by chemical reduction of graphene oxide synthesized by Hummers’ method. Reproduced with the permission of Reference [98]. Copyright 2012, Royal Society of Chemistry. (**b**) SEM images of reduced graphene oxide-P (rGO-P) bilayer film deposited on undoped Si wafer. (**c**) High-resolution C 1s spectra of rGO-P bilayer film. (**d**) Ultraviolet–visible (UV−vis) transmittance spectra of GP bilayer film (I) and rGO-P bilayer film (II). Reproduced with the permission of Reference [43]. Copyright 2018, American Chemical Society.

**Figure 5 micromachines-10-00013-f005:**
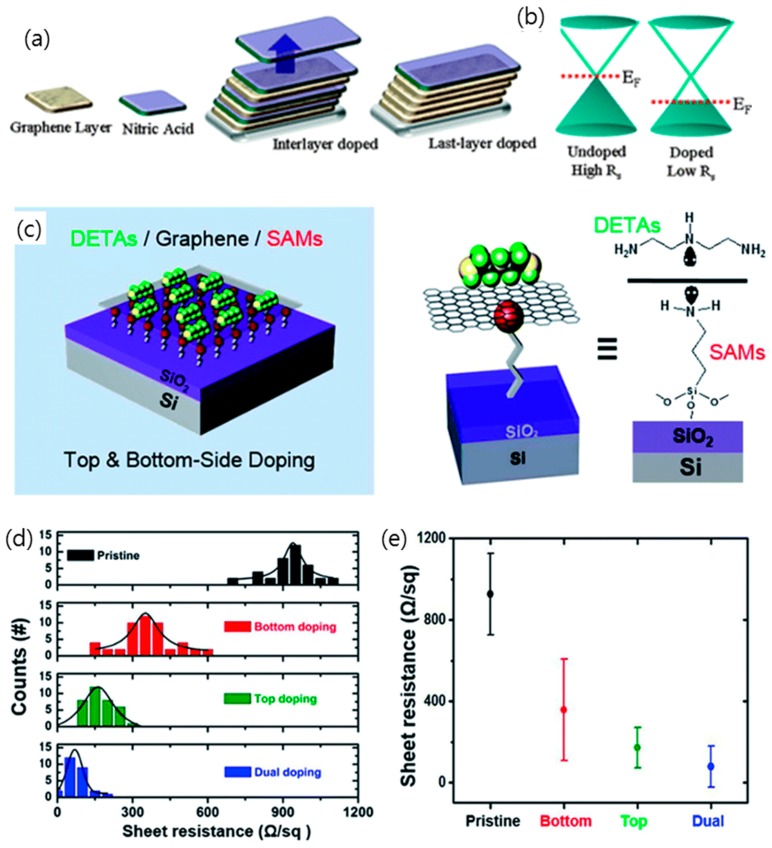
(**a**) Schematic illustrating the interlayer doping methods. The sample is exposed to nitric acid after each layer is stacked. (**b**) Illustration of the graphene band structure, showing the change in the Fermi level due to chemical p-type doping. Reproduced with the permission of Reference [105]. Copyright 2010, American Chemical Society. (**c**) Dual-side doped graphene (**left**) and graphical representation of the molecular structure of the dopants on the both sides of graphene (**right**). (**d**) Histogram of the sheet resistance of graphene doped by NH_2_-SAMs, diethylenetriamine (DETA), and DETA/NH_2_-SAMs (dual-side doped). (**e**) Averages and distributions of the sheet resistance plot of four different types of graphene field-effect transistors (FETs). Reproduced with the permission of Reference [41]. Copyright 2014, Royal Society of Chemistry.

**Figure 6 micromachines-10-00013-f006:**
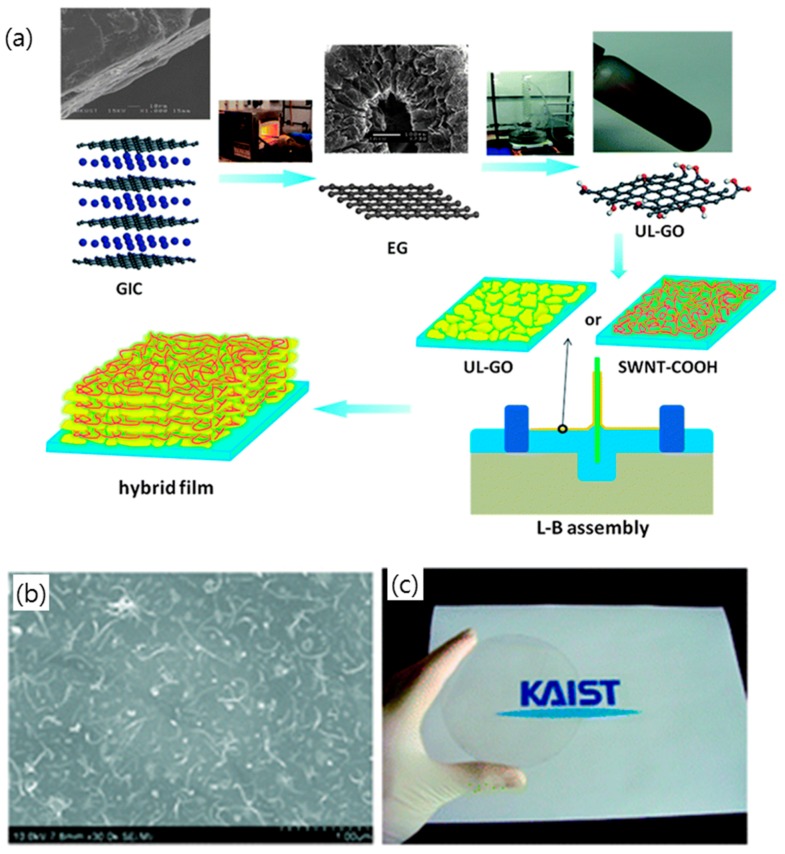
(**a**) Flow chart for the synthesis of ultra-large GO (UL-GO)/SWNT hybrid films using Langmuir–Blodgett (LB) assembly. Reproduced with the permission of Reference [50]. Copyright 2012, Royal Society of Chemistry. (**b**) SEM images of the graphene oxide multi-walled carbon nanotubes (GO/MWNT) double on 500 nm SiO_2_/Si substrates that were pretreated with 10 Mm 3-aminopropyltriethoxysilane. (**c**) A photograph of a large, transparent rGO/MWNT electrode fabricated on a 4 in quartz wafer. Reproduced with the permission of Reference [48]. Copyright 2009, American Chemical Society.

**Figure 7 micromachines-10-00013-f007:**
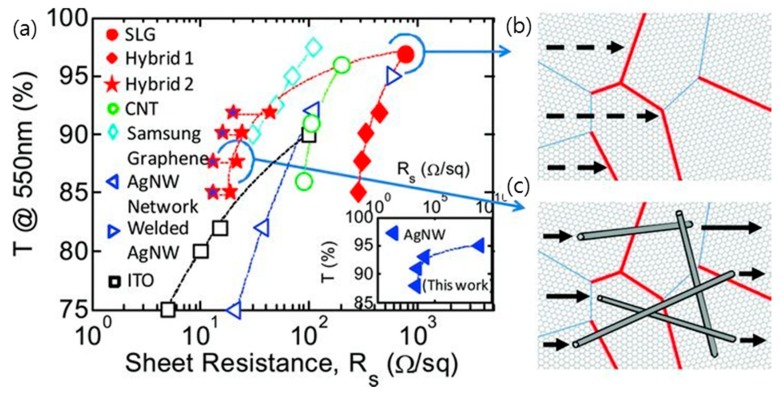
(**a**) Optical transmittance at wavelength of 550 nm (T@550nm) vs. sheet resistance (R_S_) for previous experimental reports, including networks of carbon nanotubes, networks of silver nanowires (AgNW Network and Welded AgNWs), indium tin oxide (ITO), CVD-grown single-layer polycrystalline graphene (SLG) and graphene/NW hybrid structures with various nanowire densities. (**b**,**c**) illustrate the transport across grain boundaries (GBs) in CVD SLG and hybrid SLG AgNWs networks, respectively. Low-resistance grain boundaries (LGBs, blue lines) and high-resistance grain boundaries (HGBs, red lines) are shown. The HGBs dominate the resistance in SLG. In hybrid structures with appropriate densities of AgNWs, the NWs bridge the HGBs, providing a percolating transport path for the electrons and therefore lowering the sheet resist. Reproduced with the permission of Reference [51]. Copyright 2013, Wiley-VCH.

**Figure 8 micromachines-10-00013-f008:**
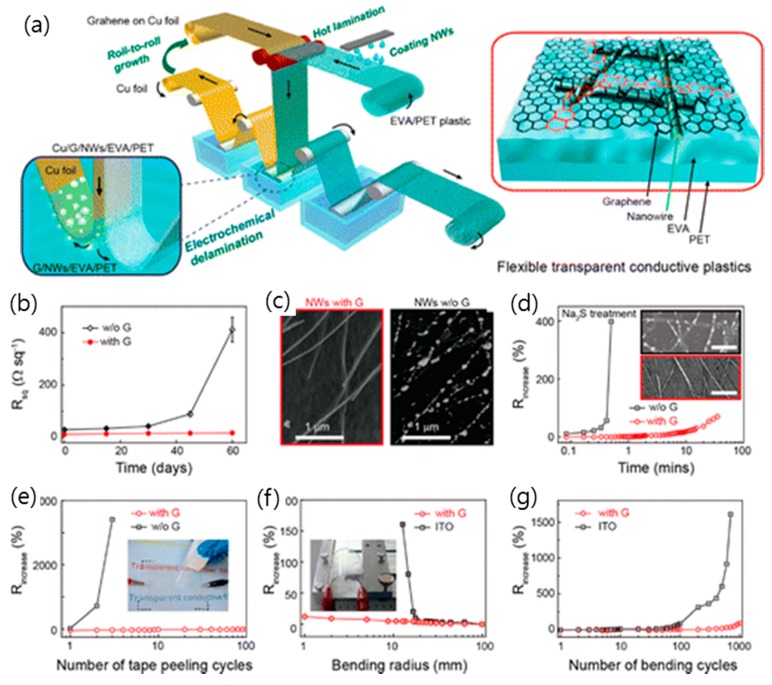
(**a**) Schematic and structure of graphene and metal nanowire hybrid films produced by a continuous roll-to-roll process. (**b**) The durability of graphene and metal nanowire hybrid transparent electrodes. Changes in sheet resistance of pure AgNW films and the graphene/AgNW hybrid films exposed in air at room temperature for 2 months. (**c**) SEM image of the graphene/AgNW hybrid film and pure AgNW films exposed in air for two months, revealing that AgNWs without the protection of graphene were oxidized to break. (**d**) Changes in sheet resistance of pure AgNW films and the graphene/AgNW hybrid films under the attack of aqueous Na_2_S (4 wt.%). (Inset) Morphologies of AgNWs with or without the graphene coverage attacked for 30 s, respectively. Scale bar: 1 μm. (**e**) Variations in sheet resistance of pure AgNW films and graphene/AgNW hybrid films as a function of the number of cycles of repeated peeling by 3M Scotch tape. (**f**) Variations in sheet resistance versus bending radius for the hybrid transparent plastic electrodes and ITO films on 150 μm thick PET. (**g**) Variations in sheet resistance of the hybrid transparent plastic electrodes and ITO films on PET as a function of the number of cycles of repeated bending to a radius of 20 mm. Reproduced with the permission of Reference [54]. Copyright 2015, American Chemical Society.

**Figure 9 micromachines-10-00013-f009:**
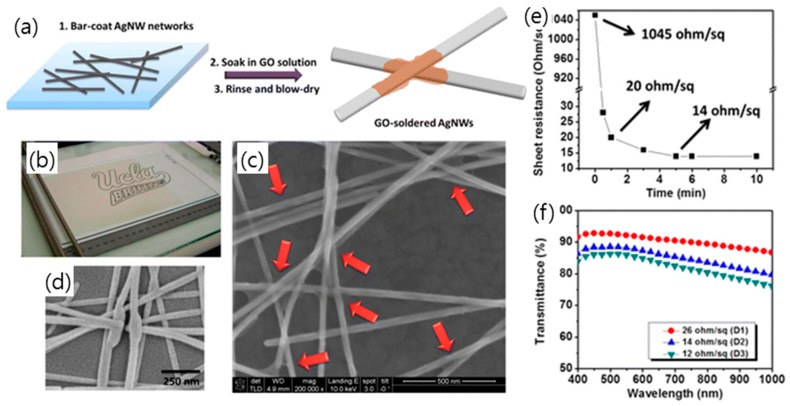
(**a**) Schematic illustration of the fabrication of a GO-AgNW network on a glass substrate at room temperature. (**b**) Optical photograph of a AgNW network bar-coated on a drawdown machine. SEM images of (**c**) GO-soldered AgNW junctions (indicated by red arrows) and (**d**) typical high-temperature fused AgNW junctions. (**e**) Measured sheet resistance of a GO-AgNW network as a function of the soaking time in a GO solution. (**f**) Transmittance spectra of GO-AgNW networks with three different AgNW densities (D1, D2, and D3). Reproduced with the permission of Reference [130]. Copyright 2014, American Chemical Society.

**Figure 10 micromachines-10-00013-f010:**
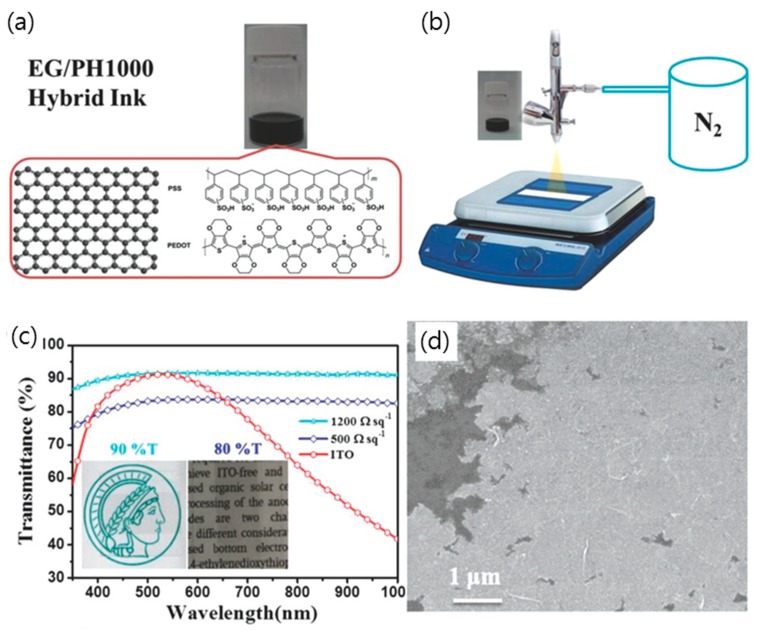
(**a**) Digital image of EG/PH1000 hybrid ink; molecular structures of EG and PH1000. (**b**) Schematic illustration of spray-coating an EG/PH1000 hybrid ink onto desired substrates. (**c**) Transmittance spectrum of both EG/PH1000 hybrid films and ITO on PET substrates. Inset shows the optical images of the EG/PH1000 hybrid films on PET substrates with 90% and 80% transmittance, respectively. (**d**) SEM image of spray-coated EG/PH1000 hybrid film. Reproduced with the permission of Reference [58]. Copyright 2015, Wiley-VCH.

**Figure 11 micromachines-10-00013-f011:**
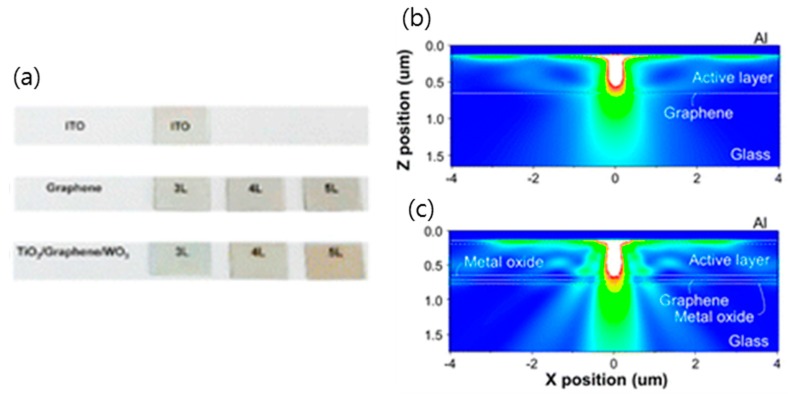
(**a**) Photographs of ITO, graphene, TiO_2_/graphene/WO_3_ of 3–5 graphene layers. The Finite-Difference Time-Domain (FDTD) simulation. Calculated Poynting vector magnitude of OLEDs (λ = 520 nm) on (**b**) 3 layers graphene (G) and (**c**) TiO_2_/3 layers graphene/WO_3_ electrodes by FDTD method. Reproduced with the permission of Reference [69]. Copyright 2016, American Chemical Society.

**Figure 12 micromachines-10-00013-f012:**
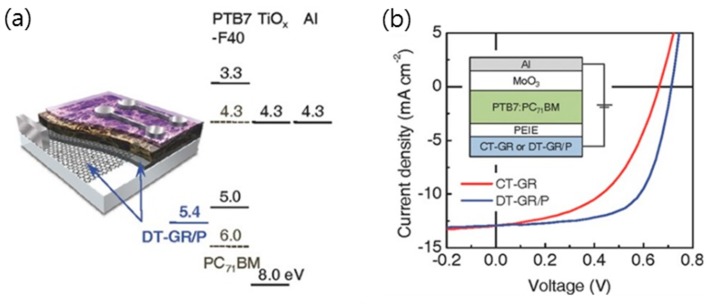
(**a**) Schematic illustrating the structure and corresponding energy-level diagram of the fabricated conventional structure perovskite solar cells (PSC). For some of the devices, an additional PEDOT:PSS layer (AI 4083, 30 nm) was used with the doping transfer-graphene (DT-GR)/P electrode. (**b**) The current density-voltage (J–V) characteristics of the best inverted structure PSCs with the conventional transfer (CT)-GR (red line) and the DT-GR/P (blue line) electrodes. The inset represents the device structure of the inverted PSCs. Reproduced with the permission of Reference [60]. Copyright 2014, Wiley-VCH.

**Figure 13 micromachines-10-00013-f013:**
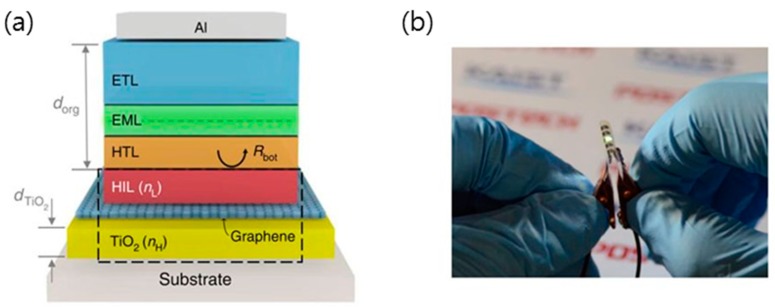
(**a**) Schematic device structure of the proposed flexible TiO_2_/graphene Organic light-emitting diodes (OLEDs). (**b**) Photograph of the proposed OLEDs in operation. Reproduced with the permission of Reference [139]. Copyright 2016, Nature Publishing Group.

**Figure 14 micromachines-10-00013-f014:**
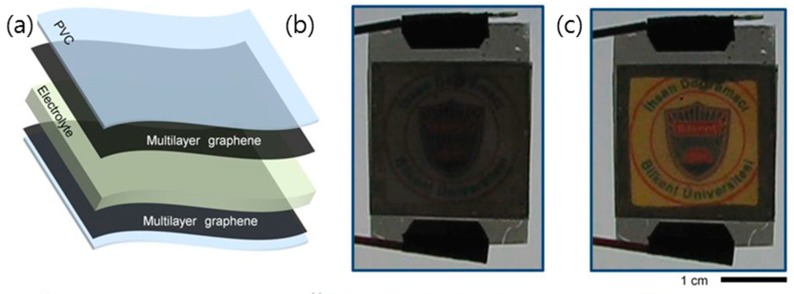
(**a**) Exploded-view illustration of the graphene electrochromic device. The device is formed by attaching two graphene coated polyvinyl chloride (PVC) substrates face to face and imposing ionic liquid in the gap separating the graphene electrodes. (**b**,**c**), Photographs of the devices under applied bias voltages of 0 V and 5 V, respectively. Reproduced with the permission of Reference [140]. Copyright 2014, Nature Publishing Group.

**Figure 15 micromachines-10-00013-f015:**
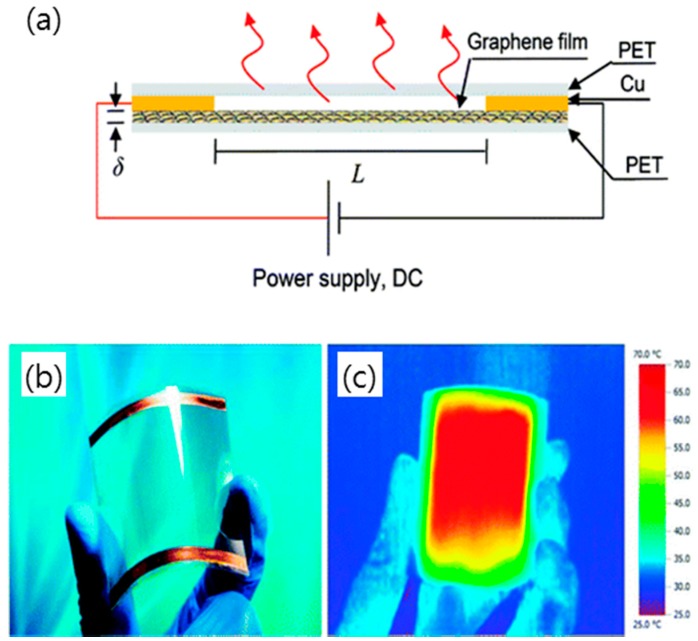
(**a**) A schematic structure of a transparent, flexible graphene heater combined with a plastic substrate and Cu electrodes. (**b**) An optical image of the assembled graphene-based heater showing its outstanding flexibility. (**c**) An infrared picture of the assembled graphene-based heater while applying an input voltage under bending condition. Reproduced with the permission of Reference [143]. Copyright 2011, American Chemical Society.

**Table 1 micromachines-10-00013-t001:** The performance of transparent conducting electrodes (TCEs) based on graphene-related materials. CVD–chemical vapor deposition, DETA–diethylenetriamine, rGO–reduced graphene oxide, CNT–carbon nanotubes, SWNT–single-walled carbon nanotubes, MWNT–multi-walled carbon nanotubes, NW–nanowire, GP–Graphene, PEDOT:PSS–poly(3,4-ethylenedioxythiophene) polystyrene sulfonate, ITO–indium tin oxide.

Material	Details	Deposition/Transfer Techniques	Sheet Resistance (Ω∙sq^−1^)	Transmission (%)	Ref.
CVD graphene	HNO_3_ doping	Dry transfer/thermal release tape	~30 (4-layers)	90	[12]
	Cu catayst	Polymer-free transfer	810 (1-layer) 230 (4-layers)	97.4 (1-layer) 89.4 (4-layers)	[33]
	Cu catalyst, HNO_3_ doping	Clean-lifting transfer	50 (4-layers)	~90 (4-layers)	[34]
	Cu catalyst	Roll-to-Roll green transfer	97.5	5.2k	[35]
	Ni catalyst	Wet transfer	500	75	[28]
	No catalyst	Direct CVD	370–510	82	[36]
	No catalyst, 400–600 °C	Direct CVD	5.2k	84.6	[37]
	Ni/C films on dielectrics	Transfer-free growth	50	96	[38]
	Cu-Ni alloy	Wet transfer	409	96.7	[39]
	Layer-by-layer, acid-doping	Wet transfer	80 (4-layers)	90 (4-layers)	[40]
	Dual n-doping (NH_2_-SAMs/DETA)	Wet transfer	86 ± 39	96	[41]
rGO	Theraml reduction of GO	Spin -coating	10^2^–10^3^	80	[42]
	rGO/POEGMA layer	Dip-coating	23.8k	90	[43]
	Thermal reduction of GO	Filtration	43k	95	[44]
	Thermal reduction of GO	Dip-coating	1.8 ± 0.08k	70.7	[45]
Graphene/CNT	Graphene growh on SWNT	Wet transfer	300	96.4	[46]
	Graphene flake/SWNT	Filtration	100	80	[47]
	CVD Synthesis	Wet transfer	~600	95.8	[48]
	Thermal reduction of rGO on MWNT	Electrostatic adsorption	151k	93	[48]
	Chemically converted grpahene/SWNT hybrid suspension	Spin-coating	636	92	[49]
	Ultralarge GO/SWNT	Langmuir-Blodgett	180–560	77–86	[50]
Graphan/metallic nano-structure	CVD graphene on AgNW	-	22	88	[51]
	AgNW on GP	-	33	94	[52]
	GP on AgNW	-	64 ± 6.1	93.6	[53]
	Roll-to-roll encapsulation	-	8	94	[54]
	Graphene/CuNW -Core/shell structure	-	36	79	[55]
	Graphene/CuNW Embedded structure	-	25	82	[56]
	Ag-mesh/Graphene	-	5.39 (GP on mesh) 4.54 (Mesh on GP)	88.1 (GP on mesh) 89.3 (Mesh on GP)	[57]
Graphene/organics	Graphene/PEDOT:PSS, hybrid ink	Spray coating	600	80	[58]
	rGO/PEDOT:PSS, hybrid ink	Filtration	2.3k	80	[59]
	PEDOT:PSS supproting layer on CVD graphene	Wet transfer	80 ± 4	84.6	[60]
Graphene/inorganics	CVD graphene on ITO film	-	76.46	88.25	[61]
	ITO nanoparticle on CVD graphene	-	522.21	85	[62]

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
