# Peer review of "Transparent Conductive Electrodes Based on Graphene-Related Materials"

_micromachines, 2018, doi:10.3390/mi10010013_

Round 1

Reviewer 1 Report

I commend the author on such a broad manuscript.  It was quite an undertaking to distill so much knowledge so well.  

My main suggestions are to improve the resolution of figures 1 and 15c.  While the images are crisp some of the legends need improvement.  The Scale in Figure 15 c is hard to read and need to be bigger.  In figure 1, the numbers in the colored squares in the bottom right hand corner are hard to read.   

Other than that, it is written well and I found it an easy read that was dense with good information.  

Author Response

Response to Reviewer 1 Comments

The author appreciates the useful comments provided by the reviewers. The reviewers’ comments and suggestions have greatly helped enrich the manuscript. Responses to the reviewers’ comments are summarized below:

Point 1: My main suggestions are to improve the resolution of figures 1 and 15c.  While the images are crisp some of the legends need improvement.  The Scale in Figure 15 c is hard to read and need to be bigger.  In figure 1, the numbers in the colored squares in the bottom right hand corner are hard to read. 

Response 1: First, I downloaded figure 1 and 15c again and edited the figures to make them look better. In figure 1, the colors of 6, 7 numbers are made white so that they are clearly visible. And, in Figure 15c, the size of the picture is slightly larger so that the scale can be seen more clearly.

Reviewer 2 Report

The review manuscript overviews development of transparent conductive electrodes based on graphene related materials. The manuscript is well written, clear and interesting. It has just few minor points to be improved.

Similar review, where electrodes based carbon and metallic materials were overviewed, was      published in 2017 and it is not cited. Luo et al. Micromachines 2017, 8, 12.

Manuscript is prepared by one author, but it is written in plural (“we”). Passive is more suitable in this case.

Table 1 is not cited in the manuscript. Unclear, why it is not in the text, but at the end of Summary. It is better to incorporate into the main text.

Author Response

Response to Reviewer 2 Comments

The author appreciates the useful comments provided by the reviewers. The reviewers’ comments and suggestions have greatly helped enrich the manuscript. Responses to the reviewers’ comments are summarized below:

Point 1: Similar review, where electrodes based carbon and metallic materials were overviewed, was published in 2017 and it is not cited. Luo et al. Micromachines 2017, 8, 12.

Response 1: I added Luo et al.'s article you mentioned in reference 4 as follows.

4. Luo, M.; Liu, Y.; Huang, W.; Qiao, W.; Zhou, Y.; Ye, Y.; Chen, L.-S. Towards flexible transparent electrodes based on carbon and metallic materials. Micromachines 2017, 8, 12

Point 2: Manuscript is prepared by one author, but it is written in plural (“we”). Passive is more suitable in this case.

Response 2:  I revised the sentence on line 57 of page 2 to read as follows:

In this paper, we will review the fabrication method of TCEs using graphene or GO (or rGO), which have been studied previously, and the optical, electrical and mechanical properties with their limited application” is revised to “This paper will review the fabrication methods of TCEs using graphene or GO (or rGO) which have been studied previously, and the optical, electrical, and mechanical properties with their limited application”

Point 3: Table 1 is not cited in the manuscript. Unclear, why it is not in the text, but at the end of Summary. It is better to incorporate into the main text.

Response 3: I cited Table 1 on page 2, line 62 of the main text, and placed Table 1 on pages 2 and 3.

Line 62 of the page 2 :The performance of various hybrid TCEs based on graphene are summarized in Table 1.”
